# A Highly Sensitive and Selective Nano-Fluorescent Probe for Ratiometric and Visual Detection of Oxytetracycline Benefiting from Dual Roles of Nitrogen-Doped Carbon Dots

**DOI:** 10.3390/nano12234306

**Published:** 2022-12-04

**Authors:** Huifang Wu, Mengqi Xu, Yubing Chen, Haoliang Zhang, Yongjun Shen, Yanfeng Tang

**Affiliations:** Nantong Key Lab of Intelligent and New Energy Materials, School of Chemistry and Chemical Engineering, Nantong University, Nantong 226019, China

**Keywords:** oxytetracycline, nitrogen-doped carbon dots, lanthanide, ratiometric, visual analysis

## Abstract

The specific detection of oxytetracycline (OTC) residues is significant for food safety and environmental monitoring. However, rapid specific determination of OTC from various tetracyclines is still challenging due to their similar chemical structures. Here, nitrogen-doped carbon dots (NCDs) with excitation and pH-dependent optical properties and a high-fluorescence quantum yield were successfully synthesized, which were directly employed to fabricate a dual-response fluorescence probe by self-assembly with Eu^3+^ (NCDs/Eu^3+^) for the ratiometric determination of OTC. The addition of OTC into the probe greatly enhances the characteristic emission of Eu^3+^ due to the “antenna effect”, and the incorporation of NCDs into the probe further improves the Eu^3+^ fluorescence by remarkably weakening the quenching effect caused by H_2_O molecules and efficiently shortening the distance of energy transfer from OTC to Eu^3+^. Meanwhile, the fluorescence of NCDs apparently decreases due to aggregation-caused quenching. The results demonstrate that a ratiometric detection of OTC (0.1–25 µM) with a detection limit of 29 nM based on the double response signals is achieved. Additionally, visual semi-quantitative assay of OTC can be realized with the naked eye under a 365 nm UV lamp according to the fluorescence color change of the as-fabricated probe. This probe exhibits acceptable specificity and anti-interference for OTC assay, holding promise for the fast detection of OTC in real water and milk samples.

## 1. Introduction

Tetracyclines (TCs) are a class of inexpensive, broad-spectrum antibacterial agents that are widely used in aquaculture and animal husbandry [1]. Oxytetracycline (OTC), as a member of TCs, is most commonly used as a therapeutic agent in food-producing animals to prevent common and respiratory infections due to its antibacterial properties against both Gram-negative and Gram-positive bacteria [2,3]. In addition, OTC can act as a growth promoter to improve animal weight gain and increase milk production of dairy animals [4]. However, excessive use of OTC severely leads to large residues in animal-derived foods, which poses a serious threat to human health, as it is harmful to consumers’ liver and skeleton and increases the antibiotic resistance of pathogenic bacteria [5,6]. Moreover, OTC residues are nearly persistent, which can accumulate continuously and enter the ecological cycle including ground water, surface water and drinking water [7]. Therefore, the monitoring of TC residues such as OTC in animal foods and aquatic environments is of significance for human health. Traditional analytical techniques for the determination of residual levels of OTC include high-performance liquid chromatography [8], capillary electrophoresis [9], liquid chromatography tandem mass spectrometry [10] and enzyme-linked immunosorbent assay [11], which suffer from a tedious and time-consuming sample preparation process, precise laboratory equipment and technical expertise [12]. Therefore, it is urgent to develop low-cost approaches for the fast detection of OTC with high sensitivity and selectivity.

To date, colorimetric, electrochemical and fluorescence-based sensing methods are widely used to determine multiple analytes in environmental and biological samples [13,14]. In particular, fluorescence assays with short response time, easy operation, high sensitivity, low cost and visualization have attracted much attention [15]. With the rapid development of functional materials, a large number of nanomaterials with distinctive optical properties, such as conjugated polymers, metal nanoclusters, semiconductor quantum dots, graphene quantum dots and carbon dots (CDs), have been applied to fabricate fluorescence sensing nanoplatforms for target detection [16,17,18,19]. Among them, CDs are more attractive because of their simple eco-friendly manufacturing process, excellent photostability and water dispersibility, low toxicity and biocompatibility, as well as tunable emission wavelength, which show potential in fluorescence measurement [20,21]. Several CD-based fluorescence assays have been developed for the detection of OTC [22,23,24]. For example, Wang et al. [22] prepared piperazine-modified carbon dots (P-CDs) for OTC assay based on the OTC-induced red shift of the emission peak that tends to be interfered by probe concentration, instrument conditions, environmental factors, etc., due to its single-emission signal. Ratiometric detection based on double-emission signals can effectively reduce external interference through self-calibration, and thereby remarkably improves detection sensitivity and accuracy [25]. Furthermore, the distinct fluorescence color variation of the probe upon incubation with analytes can be easily distinguished by the naked eye, which largely benefits rapid visual analysis in daily life [14,17]. Song et al. [23] developed CDs for OTC detection based on Förster resonance energy transfer (FRET), realizing the fast ratiometric determination of OTC. However, the development of CDs with dual response is still challenging and it is worth exploring for next-generation sensing applications.

Recently, ratiometric probes have been fabricated by integrating CDs with lanthanideions for TC determination [26,27,28]. Lanthanide complex-based probes possess many attractive advantages, such as low biotoxicity, high stability, large Stokes shift, line-shaped emission peaks, long fluorescence lifetime as well as resistance to photobleaching, endowing them significant potential for ratiometric analysis, visual inspection and bioimaging [29,30]. Wang et al. [26] designed a double-emitting fluorescent probe for the selective detection of three TCs on the basis of boron nitride-doped CDs and Eu^3+^ ion hybrid. Despite the development of fluorescent probes for TC detection, the specific recognition of OTC from multiple TCs is still difficult because of their similar structures. Therefore, it is of great urgency to achieve sensitive and selective determination of OTC.

In this work, a novel ratiometric fluorescence probe is developed by integrating nitrogen-doped CDs and europium ions (NCDs/Eu^3+^) for selective analysis of OTC. As depicted in Figure 1, the NCDs were facilely synthesized by a hydrothermal process using citric acid and uric acid as starting materials, which could emit strong blue fluorescence with a quantum yield of 15.48%. The NCDs/Eu^3+^ probe fabricated by self-assembly with NCDs as ligands and Eu^3+^ ions as metal notes shows NCD fluorescence at only 438 nm. With the introduction of OTC, the emission of NCDs was significantly quenched because of aggregation-caused quenching (ACQ). The characteristic fluorescence of Eu^3+^ increased, attributed to “antenna effect”, and NCDs acting as ligands of Eu^3+^ can further enhance Eu^3+^ fluorescence by efficiently eliminating the quenching effect caused by H_2_O molecules and greatly shortening the distance of energy transfer from OTC to Eu^3+^. Based on the double response signals, this novel probe exhibited superior selectivity for OTC compared to other TCs or antibiotics, and a detection limit of 29 nM was obtained. More importantly, the clear blue fluorescence turning to purple for rapid visual semi-quantification of OTC was easily realized, and the applicability of the probe in water and milk samples was realized with satisfactory detection results.

## 2. Materials and Methods

### 2.1. Materials and Structural Characterization

Citric acid (CA), uric acid (UA), europium nitrate hexahydrate (Eu(NO_3_)_3_·6H_2_O), nicotinic acid, glucose, D-alanine, L-glutamic acid, L-cysteine and L-leucine were purchased from Aladdin Chemistry Co., Ltd. (Shanghai, China). Antibiotics including oxytetracycline, tetracycline, chlortetracycline, doxycycline, metacycline, minocycline, penicillin, azithromycin, streptomycin, ciprofloxacin and norfloxacin were purchased from Shanghai Macklin Biochemical Co., Ltd. (Shanghai, China). All common metal salts were purchased from Sinopharm Chemical Regents Co., Ltd. (Shanghai, China). HEPES buffer (200 mM) with different pH values was prepared by titrating 200 mM HEPES aqueous solution with 3 M NaOH standard solution. Deionized water (18.2 MΩ cm) prepared from the UPHW-I-90T ultrapure water system was used to prepare standard solutions throughout the experiments. Apparatus and characterization are described in the Appendix A.

### 2.2. Preparation of NCDs

NCDs were prepared through a one-step hydrothermal process, as we previously reported [31]. In brief, citric acid (1 mmol, 192.1 mg) and uric acid (0.5 mmol, 84.1 mg) were placed into 10 mL of deionized water. After 10 min of ultrasonic treatment, the mixed solution was transferred into a Teflon-lined autoclave. After heating at 180 ℃ for 8 h, the reaction was stopped and a yellow NCDs dispersion was obtained. The product solution was subjected to purification through a 0.22 μm filter membrane and then filtered in a dialysis bag (1000 Da) for 24 h. The purified solution was lyophilized to obtain NCD powder. NCDs fabricated by citric acid and uric acid with other molar ratios (5:1, 1:1, 1:2, 1:5) were also obtained as mentioned above.

### 2.3. Ratiometric Detection of OTC

NCD powder was re-dispersed into deionized water to obtain 0.2 g·L^−1^ NCD dispersion. Then, 100 µL 200 mM HEPES buffer (pH = 6.5) and 100 µL 0.2 g·L^−1^ NCD solution were added into a tube. Afterwards, 80 µL 500 µM Eu^3+^ was added to prepare NCDs/Eu^3+^ probe. Finally, OTC at different concentrations (0.1–25 µM) was added into the probe, and all samples were calibrated to 1 mL with deionized water. After 20 min incubation at room temperature, the fluorescence spectrum was recorded by a spectrofluorometer with an excitation wavelength of 375 nm, and the excitation and emission silts were set to 5 and 10 nm, respectively. The quantitative detection of OTC was realized with F_617_/F_438_ (fluorescence ratio of 617 and 438 nm) as a function of OTC concentration (i.e., C_OTC_). In order to examine the detection selectivity of the probe, other interferents including antibiotics, amino acids, biomolecules as well as cations and anions were added into the probe instead of OTC. Moreover, the anti-interference ability of the proposed method was investigated in the coexistence of OTC and interferents. The reported values represent the average over at least three measurements.

### 2.4. Assay of OTC in Real Samples

The applicability of the probe was demonstrated by OTC assay in real water and milk samples. The tap water was used directly without any pre-treatment. Lake water was collected from Lake Zilang in Nantong, Jiangsu, China, purified through a 0.22 μm microporous membrane before detection. Fresh milk was purchased from a supermarket and was pretreated before detection. Briefly, the milk sample was diluted 10-fold with deionized water and then 1% trichloroacetic acid was added. After sonication for 30 min, the protein collected by centrifugation was removed. Finally, the supernatant obtained was further purified through a 0.22 μm microporous membrane, and the filtrate was collected for further analysis [32]. The spiked samples were prepared by the addition of OTC at different concentrations (2, 5, 10 μM) into pretreated samples. The OTC determination in pretreated and spiked samples was performed by the aforementioned ratiometric analysis method, and three parallel experiments were performed.

## 3. Results and Discussion

### 3.1. Synthesis and Characterization of NCDs

A transparent NCD dispersion was prepared by a one-step hydrothermal method using citric acid as the carbon source, and uric acid with non-toxicity and environmental friendliness served as the precursor of the nitrogen source. To improve the fluorescence intensity of NCDs, the effect of the molar ratio of citric acid and uric acid (M_CA_/M_UA_) as raw materials was studied. As depicted in Appendix A, the emission intensity of NCDs reaches a maximum at a molar ratio of 2:1, which is employed as a component for the fabrication of a dual-response fluorescence probe.

The morphology and size of NCDs were characterized by high-resolution transmission electron microscopy (HRTEM). As depicted in Figure 1a,b, the NCDs show adequate monodispersity and uniformity with diameters in the range of 2.5–6.5 nm, and the average diameter is 4.55 ± 0.25 nm (Figure 1c). Furthermore, there is no clear lattice structure in the NCDs owing to the disordered carbon atoms, similar to the previous results [33]. The X−ray diffraction (XRD) pattern of the NCDs shows a broad peak centered at 25.4° (Figure 1d), in agreement with the HRTEM result in Figure 1b [34]. The chemical composition and functional groups of NCDs were characterized by the Fourier−transform infrared (FT−IR) spectrum (Figure 2a). The broad adsorption band centered at 3542–3100 cm^−1^ originates from the stretching vibrations of O-H and N-H [33]. The peak at 1201 cm^−1^ represents the C-OH vibration, and the two adsorption peaks at 1720 and 1672 cm^−1^ correspond to the C=O stretching [35]. Stretching vibration of C=N/C=C is observed at 1573 cm^−1^ and that of C-O/C-N appears at 1403 cm^−1^, indicating that the nitrogen in UA has been successfully doped into NCDs [31]. The FT−IR results confirm that the oxygen- and nitrogen-containing functional groups have been successfully modified on the surface of the NCDs. The elemental composition and content analysis were also performed by X−ray energy−dispersive spectroscopy (EDX). As depicted in Figure 2b, the atomic percentages (At%) of C, N and O elements in the NCDs are 52.56%, 17.18% and 30.26%, respectively. The X−ray photoelectron spectroscopy (XPS) spectrum was recorded to further investigate the elemental composition as well as the chemical bonds of NCDs (Figure 2c). Three peaks at 284.8, 400.8 and 531.3 eV in the XPS survey scan spectrum of NCDs are assigned to C1s, N1s and O1s, respectively, and the *At*% of them adequately matches the EDX result [35]. The high-resolution C1s spectrum indicates that five types of carbon bonds are present in the NCDs, namely C-C (284.13 eV), C-N (285.11 eV), C-O (286.14 eV), C=O (287.43 eV) and O-C=O (289.20 eV) (Figure 2d) [36,37]. The high−resolution N1s spectrum is fitted into three peaks at 399.37, 400.18 and 401.08 eV, which can be indexed to pyridine N, pyrrolic N and N-H, respectively (Figure 2e) [38]. Three peaks at 530.75, 531.56 and 532.62 eV deconvoluted from the high-resolution O1s spectrum can be assigned to C=O, O-H and C-OH, respectively (Figure 2f) [21]. All of the aforementioned results confirm the successful preparation of NCDs.

The optical properties of NCDs were studied by UV–Vis absorption spectrum, fluorescence spectra, fluorescence lifetime and quantum yield (Figure 3). The UV–Vis absorption of the NCDs shows clear peaks at 286 and 325 nm (Figure 3a), which can be ascribed to π–π* transitions of conjugated skeletons and n–π* transitions of surface defects caused by heteroatom doping, respectively [39]. In addition, NCDs show a broad adsorption band extending up to 400 nm, overlapping with the excitation spectrum. Strong blue fluorescence of NCDs is obtained at an excitation wavelength of 355 nm, which can be observed by the naked eye (inset in Figure 3a). Moreover, the emission spectra of NCDs depend on the excitation wavelength due to multi-luminous centers caused by multiple functional groups modified in NCDs [34,40]. When NCDs are excited with different excitation wavelengths, their fluorescence first increases and then decreases, and the emission peaks show a red shift as well (Figure 3b). To further study luminescence properties of NCDs, their fluorescence lifetime decay curve and quantum yield were recorded. As shown in Figure 3c, two lifetime components of *τ*_1_ = 1.57 ns and *τ*_2_ = 6.80 ns are obtained according to the fluorescence lifetime decay curve, and the average lifetime (*τ_ave_*) is calculated based on Equation (1) [35],
(1)$$\tau_{ave}=\left(A_{1}\tau_{1}^{2}+A_{2}\tau_{2}^{2}\right)/\left(A_{1}\tau_{1}+A_{2}\tau_{2}\right)$$
where *A_1_* and *A_2_* represent fractional contributions for the time-resolved decay lifetimes of *τ_1_* and *τ_2_*, respectively. The *τ_ave_* is calculated to be 3.17 ns. The NCDs also show a high absolute fluorescence quantum yield (FQY) of 15.48%, as recorded by a spectrofluorometer. Additionally, the luminescence of NCDs at different pH solutions was examined (Figure 3d). With increasing pH from 6.0 to 9.5, the emission intensity of NCDs increases gradually because of the protonation of functional groups of NCD, confirming that the functional groups and surface traps mainly contribute to the photoluminescence of NCDs. Moreover, the NCDs fabricated in different batches show identical physical and optical properties, indicating the stable reproducibility of NCDs (Appendix A).

### 3.2. Fabrication of Ratiometric Fluorescence Probe for OTC Detection

The suitable optical properties and stability of uniform and small NCDs employed for the direct fabrication of a fluorescence probe open up more opportunities for practical application. As seen in Figure 4 (black line), NCDs exhibit strong blue fluorescence with an emission peak at 438 nm. The abundant functional groups on the surface of NCDs can easily coordinate with Eu^3+^ ions, and thus, the NCDs/Eu^3+^ hybrid as a fluorescence probe was prepared by simple self-assembly with NCDs as ligands and Eu^3+^ as the metal note. As shown in Figure 2a, compared to NCDs, the stretching vibration of C=O in NCDs/Eu^3+^ shifts to ~1600 cm^−1^, and that of C-O/C-N shifts to ~1380 cm^−1^. The XPS survey scan of NCDs/Eu^3+^ shows two apparent peaks at 1126 and 1155 eV, which are ascribed to binding energy of Eu3d5 and Eu3d3, respectively (Figure 2c) [41]. Moreover, slight shifts in the high-resolution N1s and O1s spectra are observed (Appendix A). All the results confirm the successful modification of Eu^3+^ on the surface of NCDs. Interestingly, the incorporation of Eu^3+^ slightly affects the fluorescence of NCDs, and no noticeable Eu^3+^ fluorescence is observed either (Figure 4a, black and red line). For the fluorescence response of the probe to OTC, it is found that the fluorescence of NCDs decreases, while the fluorescence of Eu^3+^ resulting from the transitions of ^5^D_0_ to ^7^F_1_ (592 nm) and ^5^D_0_ to ^7^F_2_ (616 nm) increases upon the addition of OTC [42], accompanied by fluorescence color changing from blue to purple (Figure 4, green line). Therefore, based on the double response signals of NCDs at 438 nm and Eu^3+^ at 617 nm, the ratiometric detection of OTC can be performed by establishing the relationship of F_617_/F_438_ and the OTC concentration. Additionally, the appropriate excitation wavelength of the probe for OTC assay was investigated. There are three emission peaks in the probe after the addition of OTC, including 438, 591 and 617 nm, and the excitation spectrum at an emission wavelength of 617 nm was examined. A maximum excitation peak can be obtained at 375 nm (Appendix A), so the sensing performance of the probe for OTC determination is investigated with λ_ex_ = 375 nm.

### 3.3. Possible Sensing Mechanism of the Probe for OTC Detection

The response mechanism of the NCDs/Eu^3+^ fluorescent probe for OTC was studied in detail. The adsorption and fluorescence spectra of NCDs and NCDs/Eu^3+^ demonstrate that the incorporation of Eu^3+^ did not apparently influence the optical properties of NCDs, and NCDs cannot sensitize the characteristic fluorescence of Eu^3+^ (Figure 4a and Figure 5a). To explore the increase mechanism of Eu^3+^ fluorescence, the UV–Vis absorption spectra and emission spectra of OTC before and after the incubation with Eu^3+^ were first recorded. As depicted in Figure 5a, the absorption spectrum of OTC after the addition of Eu^3+^ shows an apparent red shift from 360 to 390 nm and the absorption intensity increases simultaneously, indicating the successful coordination of OTC with Eu^3+^. Furthermore, the fluorescence of OTC is almost completely quenched after the introduction of Eu^3+^ due to the energy transfer from OTC to Eu^3+^, which is known as the “antenna effect” (Figure 5b) [42,43]. The fluorescence lifetime decay curves of Eu^3+^ in the probe were also recorded upon incubation with OTC at different concentrations. With the increasing concentration of OTC from 1.0 to 25 μM, the fluorescence lifetime of Eu^3+^ increases from 14.86 to 102.34 μs (Figure 5c). Therefore, the enhancement of Eu^3+^ fluorescence in the probe by OTC lies in the fact that OTC effectively absorbs the UV light, then transfers energy to Eu^3+^, and thus sensitizes characteristic fluorescence of Eu^3+^.

It is interesting to note that the NCDs also play important roles in the optimization of Eu^3+^ emission. The emission state of Eu^3+^ is susceptible to the potential quencher, namely the O-H oscillator of H_2_O molecules in an aqueous solution [43]. Although OTC offers multiple coordination sites for Eu^3+^, the limited amount of OTC suggests that Eu^3+^ ions tend to coordinate with H_2_O molecules in aqueous medium. As depicted in Figure 4 (blue line and green line), the fluorescence of Eu^3+^ in the NCDs/Eu^3+^-OTC system is stronger than that in the Eu^3+^-OTC system, which is probably because NCDs coordinate with Eu^3+^ instead of H_2_O molecules, and thereby protects Eu^3+^ from the quenching effect by H_2_O molecules. To further demonstrate the efficient shielding effect of NCDs for Eu^3+^, the emission spectra of Eu^3+^-OTC and NCDs/Eu^3+^-OTC complexes in D_2_O medium were also investigated (Figure 4, purple line and orange line). Compared to that in H_2_O medium, the Eu^3+^ emission of the Eu^3+^-OTC system significantly increased in D_2_O medium due to the absence of O-H oscillator. Interestingly, the incorporation of NCDs into the Eu^3+^-OTC system in H_2_O medium also enhanced the Eu^3+^ fluorescence, suggesting that the NCDs can remarkably shield the quenching effect of H_2_O molecules by coordinating with Eu^3+^ instead of H_2_O, leading to the enhancement of Eu^3+^ fluorescence [17]. In addition, the simultaneous coordination of Eu^3+^ with NCDs and OTC resulted in a large number of aggregates of NCDs (Appendix A, b), which probably shortened the distance of energy transfer between OTC and Eu^3+^ and promoted the energy transfer from OTC to Eu^3+^, facilitating the promotion of Eu^3+^ fluorescence [41,42]. These results demonstrate that the enhancement of Eu^3+^ fluorescence is not only attributed to the “antenna effect”, but also comes from the shielding effect of NCDs through acting as the ligands of Eu^3+^ and the shortened energy distance transfer from OTC to Eu^3+^.

For the quenching process of NCDs, first, the fluorescence spectra of NCDs upon addition of OTC in the range of 0–25 μM were collected. As seen in Appendix A, OTC hardly affects the fluorescence of NCDs. Moreover, it is demonstrated that the incorporation of Eu^3+^ does not affect the fluorescence of NCDs (Figure 4). Therefore, the quenching process of NCDs happened only in the coexistence of Eu^3+^ and OTC. The TEM images of NCDs/Eu^3+^ probe before and after incubation with OTC demonstrate the significant aggregate of NCDs (Appendix A), which can result in the aggregation-caused quenching of NCD fluorescence.

Generally, fluorophores can be quenched by means of a dynamic quenching effect (DQE) or a static quenching effect (SQE) or both simultaneously. DQE is mainly due to the deactivation of the fluorophore in the excited state upon collision with the quencher, while SQE originates from formation of a nonfluorescent complex. The Stern–Volmer equation can be applied to describe both DQE and SQE mechanisms [44]. The quenching process of NCDs in the probe by OTC was characterized by the Stern–Volmer Equation (2) [44].
(2)$$\frac{F_{0}}{F}=1+K_{SV}\times C_{OTC}$$
where *F*_0_ and *F* represent the emission intensity of NCDs before and after addition of OTC, respectively, and *K_SV_* is the quenching constant. In the quenching experiments, *F_0_*/*F* is linearly related to *C_OTC_*, and the equation is *F*_0_/*F*
*=* 0.034*C_OTC_* + 1.02 (*R*^2^ = 0.9945, Appendix A). This linear relationship indicates that the Stern–Volmer equation can be successfully used to explain the response mechanism in this study, and the *K_SV_* is 0.034 μM^−1^. In addition, for DQE, the *K_SV_* follows Equation (3) [45].
(3)$$K_{SV}=K_{q}\times \tau_{0}$$
where *K_q_* represents the quenching rate constant, and *τ_0_* denotes the fluorescence lifetime of NCDs before incubation with OTC. For NCDs, *K_SV_* = 0.034 μM^−1^ and *τ_0_* = 3.17 ns. Therefore, *K_q_* is calculated to be 1.07 × 10^13^ M^−1^·s^−1^ based on Equation (3), which is much higher than the largest possible value (1.0 × 10^10^ M^−1^·s^−1^) for the quenching of fluorophore in a diffusion-controlled process [44], demonstrating that DQE is not possible for the quenching process of NCDs. To further illustrate which quenching mechanism is involved in this quenching process of NCDs, the transient fluorescence lifetime curves of NCDs at 438 nm were recorded before and after the incubation with OTC (Figure 5d). After the inclusion of OTC at different concentrations, the fluorescence lifetime of NCDs shows no obvious change, suggesting that the SQE plays a crucial role in the NCD fluorescence quenching due to the significant aggregate of NCDs. Therefore, the above analysis confirms that the quenching of NCD fluorescence is mainly resulted from the aggregation-caused static quenching effect.

### 3.4. Optimization of Experimental Conditions

In order to achieve excellent performance of the probe for OTC detection, various experimental factors were studied, including M_CA_/M_UA_ for NCDs synthesis, NCDs and Eu^3+^ concentration for the fabrication of the NCDs/Eu^3+^ probe, response time of the probe for OTC and solution pH. As shown in Appendix A, the NCDs synthesized by M_CA_/M_UA_ of 2:1 and 1:2 show a significant fluorescence response to OTC. Furthermore, with M_CA_/M_UA_ of 2:1, the emission intensity of Eu^3+^ can reach a maximum (Appendix A), so M_CA_/M_UA_ of 2:1 was chosen for the preparation of NCDs. Then, the effect of NCD concentration was examined. With increasing concentration of NCDs, the NCD fluorescence is enhanced and its quenching efficiency in the presence of OTC also gradually improves (Appendix A). Note that the maximum sensitization efficiency of Eu^3+^ fluorescence by OTC was obtained at the NCD concentration of 20 mg·L^−1^. Therefore, NCDs of 20 mg·L^−1^ were used for further tests. The effect of Eu^3+^ concentration is depicted in Appendix A, b. Under the condition of 60 μM Eu^3+^, the fluorescence quenching of NCDs is remarkable and the enhancement of Eu^3+^ fluorescence can reach a maximum upon the inclusion of OTC, so Eu^3+^ of 60 μM was chosen. Additionally, the effect of pH was investigated (Appendix A, b). In an alkaline solution, Eu^3+^ can coordinate with hydroxide, which is not conducive to coordination of OTC and Eu^3+^, thereby weakening the antenna effect with a poor emission efficiency of Eu^3+^. At pH = 7.0, the Eu^3+^ emission can reach a maximum, so pH of 7.0 was chosen as the parameter. Finally, the incubation time was optimized by recording the fluorescence ratio F_617_/F_438_ (Appendix A). Upon the addition of OTC, F_617_/F_438_ can reach a maximum after 20 min and remain stable within 60 min. In summary, M_CA_/M_UA_ of 2:1, NCDs of 20 mg·L^−1^ and Eu^3+^ of 60 μM with pH of 7.0 and an equilibration time of 20 min were chosen for OTC determination.

### 3.5. Ratiometric Detection of OTC

The quantitative determination of OTC using NCDs/Eu^3+^ as a ratiometric fluorescent probe was investigated under optimal conditions (Figure 6a,b). With the increasing concentration of OTC in the probe, the characteristic fluorescence of Eu^3+^ gradually increased, while that of NCDs simultaneously decreased (Figure 6a). More importantly, F_617_/F_438_ increased gradually with C_OTC_, and a linear relationship of F_617_/F_438_ vs. C_OTC_ (0.1–25 μM) was obtained (Figure 6b). The corresponding linear equation is F_617_/F_438_ = 0.0013 + 0.0117C_OTC_ (*R*^2^ = 0.9925), and the limit of detection based on the formula 3δ/k is 0.029 μM, which is much lower than the allowable residue of OTC (100 ng/mL) in milk proposed by the Food Safety and Standard Authority of India (FSSAI) [12]. Moreover, the sensitivity of the proposed ratiometric approach is higher or comparable to previously reported fluorescence approaches for OTC detection (Appendix A). Furthermore, when excited with a 365 nm UV lamp, the fluorescence color of the probe exhibits the apparent change from blue to purple (inset in Figure 6b). Therefore, a visual semi-quantitative assay of OTC can be achieved by directly distinguishing the change in fluorescence color with the naked eye.

For a real sample assay, complex and diverse coexisting interfering substances may interfere with the experimental results. Therefore, selectivity and anti-interference of the NCDs/Eu^3+^ probe were investigated to evaluate its application in real samples (Figure 6c–f). Under the optimal conditions, the fluorescence responses of the probe to cations, six TCs (e.g., OTC, tetracycline, chlortetracycline, doxycycline, metacycline and minocycline), other antibiotics (e.g., penicillin, azithromycin, streptomycin, ciprofloxacin and norfloxacin) and organic molecules were recorded. As shown in Figure 6c and Appendix A, except for six TCs, there is no appreciable increase in F_617_/F_438_ for the probe upon the addition of other potential interfering substances. TCs possess similar chemical structures with β-diketonate configuration, acetamides and dimethyl-ammonium groups (Figure 6d), which have strong electron donating ability and are prone to coordinating vigorously with Eu^3+^ ions. Among the six TCs, this probe showed the most significant fluorescence response to OTC and F_617_/F_438_ reached a maximum in the presence of OTC. Furthermore, only with the addition of OTC does the probe show the most obvious fluorescent color change from blue to purple, which is attributed to the fact that OTC has more hydroxyl substituents to increase its surface electron density, thus showing stronger coordination ability with Eu^3+^ than other TCs [46]. Therefore, this probe shows a satisfactory selectivity for OTC. The anti-interference tests of the probe were also performed in the coexistence of OTC and other potential interfering substances, especially some cations which are easy to associate with OTC and NCDs (Figure 6e,f). The experimental results indicate that F_617_/F_438_ shows no apparent change compared to that upon addition of OTC alone, which is ascribed to the stronger binding ability of OTC with Eu^3+^ as well as the higher coordination number of Eu^3+^ ions compared to other cations. Overall, the NCDs/Eu^3+^ probe exhibits satisfactory selectivity and anti-interference for OTC detection, indicating its promising application in real samples assays. Moreover, the NCDs prepared in different batches were used to fabricate the NCDs/Eu^3+^ probe for OTC detection, which showed the identical sensing performance for OTC, confirming the stable reproducibility of NCDs (Appendix A).

### 3.6. Assay of Real Samples

To demonstrate the practical application of the NCDs/Eu^3+^ probe, determination of OTC in real samples including lake water, tap water and milk samples was performed. As can be seen in Table 1, no OTC residues were detected by the proposed method. Therefore, the standard addition method was carried out by adding a certain concentration of OTC into the tap water and pretreated lake and milk samples, and then the accuracy and recovery of the probe for the assay of spiked samples were analyzed. Recoveries of 94.4–106.4% and relative standard deviations (RSD) of 1.0–4.9% were obtained, respectively, confirming the satisfactory accuracy and reliability of the proposed method.

## 4. Conclusions

We rationally designed and successfully fabricated an NCDs/Eu^3+^ ratiometric fluorescence probe for the sensitive and selective detection of OTC. NCDs synthesized by a one-step hydrothermal process using citric acid and uric acid showed clear excitation and pH-dependent optical properties, and their fluorescence lifetime of 3.17 ns and a high FQY of 15.48% were recorded. On one hand, the inclusion of OTC into the probe could significantly enhance Eu^3+^ fluorescence owing to the energy transfer from OTC to Eu^3+^, and NCDs protected Eu^3+^ from the quenching effect of H_2_O molecules and shortened the distance of energy transfer, leading to a further enhancement in Eu^3+^ fluorescence. On the other hand, the fluorescence of NCDs decreased owing to the aggregation-caused static fluorescence quenching. This probe showed high selectivity and anti-interference for the ratiometric detection of OTC with a detection limit of 29 nM. More importantly, the rapid visual semi-quantitation of OTC with the naked eye via differentiating the fluorescence color change of the probe was also easily realized. This work not only provides technical support for the selective assay of OTC, but also sheds light on new designs of ratiometric fluorescence probes to achieve rapid on-site monitoring of analytes in real samples.

## Data Availability

The data presented in this study are available on request from the corresponding author.

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
