# Peer review of "A Highly Sensitive and Selective Nano-Fluorescent Probe for Ratiometric and Visual Detection of Oxytetracycline Benefiting from Dual Roles of Nitrogen-Doped Carbon Dots"

_nanomaterials, 2022, doi:10.3390/nano12234306_

Round 1

Reviewer 1 Report

The manuscript entitled “A Highly Sensitive and Selective Nano-Fluorescent Probe for Ratiometric and Visual Detection of Oxytetracycline Benefiting 3 from Dual Roles of Nitrogen-Doped Carbon Dots” by Wu et al. describes the synthesis of nitrogen-doped CDs and europium ions (NCDs/Eu3+) for selective detection of OTC. This is routine work for sensing of TC compounds. The experimental section needs to be more methodic, and the discussion part should be rewritten with proper explanation. The authors should address the following points carefully.

1.      What is the role of Eu3+ in the process? Have the authors checked the effect of OTC on NCDs only?

2.      The effect of other cations on the fluorescence of NCDs should be provided.

3.      How was the attachment of Eu3+ confirmed? The Raman spectra of the NCDs with and without the Eu3+ and OTC should be provided.

4.      How do the authors explain the evolution of the peak over 580 nm to 640 nm?

5.      What did the authors mean by D2O buffer while they used HEPES buffer?

6.      The spectral analysis of OTC should be provided and compared with the NCD-Eu3+ system to highlight the interaction mechanism.

7.      Why did the emission color of the materials (Figure 6b, inset) change in spite of the emission peak position remaining unaltered (Figure 6a)?

8.      The detection mechanism should be explained properly in the light of absorption and emission of NCD, NCD-Eu3+ and OTC.

9.      The typo and English should be corrected throughout the manuscript.

Author Response

Dear Editor,

Thank you for handling our manuscript (No.: nanomaterials-2046152), and we also would like to thank our reviewer for their valuable comments. We have carefully revised the manuscript according to the reviewers' comments. The corresponding changes have been made in the text and marked in color. We hope that you and the reviewer are satisfied with our revision, and are happy to answer any further questions.

Reviewers’ comments and our responses:

The manuscript entitled “A Highly Sensitive and Selective Nano-Fluorescent Probe for Ratiometric and Visual Detection of Oxytetracycline Benefiting from Dual Roles of Nitrogen-Doped Carbon Dots” by Wu et al. describes the synthesis of nitrogen-doped CDs and europium ions (NCDs/Eu3+) for selective detection of OTC. This is routine work for sensing of TC compounds. The experimental section needs to be more methodic, and the discussion part should be rewritten with proper explanation. The authors should address the following points carefully.

Response: Thank you very much for your review and nice suggestion.

Revising issues:
1. What is the role of Eu3+ in the process? Have the authors checked the effect of OTC on NCDs only?

Response: Thank you very much for pointing this out! The Eu3+ ions can coordinate with OTC via the β-diketonate configuration and hydroxyl (J. Hazard. Mater. 2018, 342, 158-165; J. Hazard. Mater. 2021, 409, 124935). Under the excitation light, OTC effectively absorb energy and transfer the energy to Eu3+, thus sensitizing the fluorescence of Eu3+ (Figure 5a, b). The incorporation of NCDs further improves the Eu3+ fluorescence by remarkably weakening the quenching effect caused by H2O molecules and efficiently shortening the distance of energy transfer from OTC to Eu3+ (Figure 4, blue and green line). Meanwhile, the fluorescence of NCDs decreases due to aggregation-caused quenching effect (Figure S4). Therefore, based on the double and reverse fluorescence response signals of Eu3+ and NCDs, the ratiometric detection of OTC can be realized.

The effect of OTC on NCDs was checked as shown in Figure S5, OTC hardly affects the fluorescence of NCDs.

Line 302-303: “As seen in Figure S5, OTC hardly affects the fluorescence of NCDs.”

2. The effect of other cations on the fluorescence of NCDs should be provided.

Response: Thank you very much for your nice suggestion! The effect of other cations on the fluorescence of NCDs was added as shown in Figure S12, and cations show negligible interference on the fluorescence spectra of NCDs/Eu3+.

Line 390-392: “As shown in Figure 6c and Figure S12, except for six TCs, there is no appreciable increase in F617/F438 for the probe upon the addition of other potential interfering substances.”

3. How was the attachment of Eu3+ confirmed? The Raman spectra of the NCDs with and without the Eu3+ and OTC should be provided.

Response: Thank you very much for your nice suggestion! The coordination of Eu3+ with NCDs was confirmed by recording the FT-IR spectrum and XPS spectrum of NCDs before and after the addition of the Eu3+. The Raman signal of the NCDs is relatively weak, which is difficult and unreliable to be employed for interaction investigation between Eu3+ and NCDs. Moreover, the absorption spectra and fluorescence spectra of NCDs before and after the addition of OTC show no apparent change (Figure 5a and Figure S5), indicating that there is no direct interaction between NCDs and OTC.

Line 236-242: “As shown in Figure 2a, compared to NCDs, the stretching vibration of C=O in NCDs/Eu3+ shifts to ~1600 cm−1, and that of the C-O/C-N shifts to ~1380 cm−1. The XPS survey scan of NCDs/Eu3+ shows two apparent peaks at 1126 and 1155 eV, which are ascribed to binding energy of Eu3d5 and Eu3d3, respectively (Figure 2c). Moreover, the slight shift of high-resolution N 1s and O 1s spectrum is observed (Figure S2). All the results confirm the successful modification of Eu3+ on the surface of NCDs.”

4. How do the authors explain the evolution of the peak over 580 nm to 640 nm?

Response: Thank you very much for your nice suggestion! The fluorescence peak ranges from 580 to 640 nm denotes the characteristic emission of Eu3+, which is ascribed to the hypersensitive transitions from 5D0 to 7F1 (592 nm) and 5D0 to 7F2 (616 nm) in Eu3+. Upon the addition of OTC into the NCDs/Eu3+ system, the Eu3+ coordinates with OTC via the β-diketonate configuration and hydroxyl. Under the excitation light, OTC can effectively absorb energy and transfer the energy to Eu3+, leading to the electron transitions from 5D0 to 7F1 and 5D0 to 7F2, improving the fluorescence of Eu3+. Moreover, the incorporation of NCDs further improves the Eu3+ fluorescence by remarkably weakening the quenching effect caused by H2O molecules and efficiently shortening the distance of energy transfer from OTC to Eu3+.

Line 244-248: “For the fluorescence response of the probe to OTC, it is found that the fluorescence of NCDs decreases, while the fluorescence of Eu3+ resulting from the transitions of 5D0 to 7F1 (592 nm) and 5D0 to 7F2 (616 nm) increases upon the addition of OTC, accompanied by fluorescence color changing from blue to purple (Figure 4, green line).”

5. What did the authors mean by D2O buffer while they used HEPES buffer?

Response: Thank you very much for your nice suggestion! The H2O molecules in an aqueous solution can easily coordinate with the Eu3+ via the oxygen atom, while O-H oscillator of H2O molecules is a common quencher of Eu3+ fluorescence (Inorg. Chem. 2013, 52, 1476-1487; J. Hazard. Mater. 2021, 409, 124935; ACS Appl. Mater. Interfaces 2020, 12, 22593-22600). Therefore, the fluorescence of Eu3+ in Eu3+-OTC system is still very weak despite of the efficient energy transfer from OTC to Eu3+ (Figure 4a, the blue line). As shown Figure 4a (the purple line), the fluorescence of Eu3+ enhances apparently in the D2O buffer due to the absence of quenching effect caused by O-H oscillator. Interestingly, it is found that the incorporation of NCDs into Eu3+-OTC system in the aqueous solution can apparently enhance the fluorescence of Eu3+ (Figure 4a, the green line), which indicates that the NCDs can remarkably weaken the quenching effect from H2O molecules due to their coordination with Eu3+ instead of H2O molecules. Therefore, the important role of NCDs in the enhancement of Eu3+ fluorescence can be confirmed by comparing the fluorescence intensity of Eu3+ in H2O medium and D2O medium.

6. The spectral analysis of OTC should be provided and compared with the NCD/Eu3+ system to highlight the interaction mechanism.

Response: Thank you very much for your nice suggestion! The adsorption spectra and fluorescence spectra of OTC before and after the addition of Eu3+ were recorded as shown in Figure 5a, b.

Line 263-268: “As depicted in Figure 5a, the absorption spectrum of OTC after the addition of Eu3+ shows an apparent red shift from 360 to 390 nm and the absorption intensity increases simultaneously, indicating the successful coordination of OTC with Eu3+. Furthermore, the fluorescence of OTC is almost completely quenched after the introduction of Eu3+ due to the energy transfer from OTC to Eu3+, which is known as “antenna effect” (Figure 5b).”

Line 278-282:“As depicted in Figure 4 (blue line and green line), the fluorescence of Eu3+ in the NCDs/Eu3+-OTC system is stronger than that in the Eu3+-OTC system, which is probably due to that NCDs coordinate with Eu3+ instead of H2O molecules, and thereby protects Eu3+ from quenching effect by H2O molecules.”

7. Why did the emission color of the materials (Figure 6b, inset) change in spite of the emission peak position remaining unaltered (Figure 6a)?

Response: Thank you very much for pointing this out! Before the addition of OTC, the NCDs/Eu3+ probe shows only strong blue fluorescence with an emission peak at 438 nm. Upon the addition of OTC into the probe, the characteristic emission of Eu3+ at 617 nm enhances gradually due to the “antenna effect”, which emits strong red fluorescence. However, the blue fluorescence of NCDs gradually decreases due to the aggregation-caused quenching. Therefore, it is concluded that the double and reverse fluorescence signals at 438 and 617 nm lead to the change of emission color of the probe.

8. The detection mechanism should be explained properly in the light of absorption and emission of NCD, NCD/Eu3+ and OTC.

Response: Thank you very much for pointing this out! The adsorption and emission spectra of different systems were recorded as shown in Figure 4a and Figure 5a.

Line 258-268: “The adsorption and fluorescence spectra of NCDs and NCDs/Eu3+ demonstrate that the incorporation of Eu3+ did not influence the optical properties of NCDs apparently, and NCDs can not sensitize the characteristic fluorescence of Eu3+ (Figure 4a and Figure 5a). To explore the increasement mechanism of Eu3+ fluorescence, the UV-Vis absorption spectra and emission spectra of OTC before and after the incubation with Eu3+ were first recorded. As depicted in Figure 5a, the absorption spectrum of OTC after the addition of Eu3+ shows an apparent red shift from 360 to 390 nm and the absorption intensity increases simultaneously, indicating the successful coordination of OTC with Eu3+. Furthermore, the fluorescence of OTC is almost completely quenched after the introduction of Eu3+ due to the energy transfer from OTC to Eu3+, which is known as “antenna effect” (Figure 5b)”

9. The typo and English should be corrected throughout the manuscript.

Response: Thanks for your kind reminder. The typo and English have been revised carefully.

Thank you again for your considerations!

Sincerely,

Yours,

Dr. Huifang Wu

Email: whf0125@ntu.edu.cn

Nantong Key Laboratory of Intelligent and New Energy Materials, Nantong University, Nantong 226019, Jiangsu, China

Nov. 28th, 2022

Reviewer 2 Report

Manuscript of Huifang Wu et al. devoted to the development of fluorescent probe for ratiometric detection of oxytetracycline. The authors use the ration of two emission wavelengths – one from NCD emission and another one from Eu emission.

The manuscript is well written and can be published after minor revision.

The main question is related to the reproducibility of NCD physical and analytical properties from synthesis to synthesis. This data should be added to the manuscript.

The second question is related to the amount of significant digits.

What is the error in determining the quantum yield? Four significant digits (15.48%) is clearly more than the measurement and calculation error allows.

In table 1 detected concentration is presented as 19.76±0.53. What is the error of determination of concentration? How were these data calculated? Some неудачных phrases it is better to re-write. Such as:

Line 12 - nitrogen-doped carbon dots (NCDs) with excitation wavelength,

Line 142 - Tap water was taken from domestic water,

Line 371 - other molecules, unions and cations

Author Response

Dear Editor,

Thank you for handling our manuscript (No.: nanomaterials-2046152), and we also would like to thank our reviewer for their valuable comments. We have carefully revised the manuscript according to the reviewers' comments. The corresponding changes have been made in the text and marked in color. We hope that you and the reviewer are satisfied with our revision, and are happy to answer any further questions.

Reviewers’ comments and our responses:

Manuscript of Huifang Wu et al. devoted to the development of fluorescent probe for ratiometric detection of oxytetracycline. The authors use the ratio of two emission wavelengths – one from NCD emission and another one from Eu emission.

The manuscript is well written and can be published after minor revision.

Response: Thank you very much for your positive response. 

Revising issues:

1. The main question is related to the reproducibility of NCDs physical and analytical properties from synthesis to synthesis. This data should be added to the manuscript.

Response: Thank you very much for your nice suggestion! For the reproducibility of NCDs, we compared the physical and analytical properties of NCDs prepared in three batches (NCDs 1, NCDs 2, NCDs 3), and the results were shown in Table S1 and Table S2. The physical and analytical properties of NCDs show satisfactory reproducibility, indicating the reliability and promising application of the proposed method.

Line 220-221: “Moreover, the NCDs fabricated in different batches show the identical physical and optical properties, indicating the stable reproducibility of NCDs (Table S1).”

Line 408-411: “Moreover, the NCDs prepared in different batches were used to fabricate the NCDs/Eu3+ probe for OTC detection, which showed the identical sensing performance for OTC, confirming the stable reproducibility of NCDs (Table S2)”

Table S1. The reproducibility of NCD physical and optical properties.

NCDs

Particle size (nm)

Fluorescence quantum yield

(%)

Fluorescence lifetime

(ns)

Excitation wavelength- and pH-dependent optical property

NCDs 1

4.89 ± 0.37

16.10

4.01

yes

NCDs 2

4.02 ± 0.30

14.95

3.03

yes

NCDs 3

4.55 ± 0.25

15.48

3.17

yes

Table S2. The reproducibility of NCDs analytical properties for OTC detection.

NCDs

Quantitative range for OTC

LOD for OTC

Selectivity for OTC

NCDs 1

0.1-25 µM

40 nM

yes

NCDs 2

0.1-25 µM

18 nM

yes

NCDs 3

0.1-25 µM

29 nM

yes

2. The second question is related to the amount of significant digits. What is the error in determining the quantum yield? Four significant digits (15.48%) is clearly more than the measurement and calculation error allows. In table 1 detected concentration is presented as 19.76±0.53. What is the error of determination of concentration? How were these data calculated?

Response:

The absolute fluorescence quantum yields of the NCDs were measured by employing a standard barium sulfate coated integrating sphere (150 mm in diameter, Edinburgh) as the sample chamber that was mounted on the FLS980 spectrometer with the entry and output port of the sphere located in 90° geometry from each other in the plane of the spectrometer, a standard tungsten lamp was used to correct the optical response of the instrument. The absolute fluorescence quantum yields can be calculated directly by the software of the FLS980 spectrometer.

For the OTC assay with the proposed method in real samples, since no OTC residues were detected in real samples, the spiked samples were analyzed and three parallel experiments were carried out. Take the concentration of 19.76±0.53 mM detected in lake water as an example. First, lake water was spiked with 20 mM OTC to prepare the spiked sample; then the spiked sample was added to the probe and the fluorescence spectra of the mixed system were recorded, three parallel experiments were performed to reduce the experimental error; the fluorescence ratio, F617/F438, of three parallel experiments can be calculated to be 0.23, 0.24, and 0.232, respectively, and the concentration of OTC (COTC) was calculated as 19.36, 20.14, and 19.77 mM, respectively, based on the established linear equation, F617/F438 = 0.0013 + 0.0117COTC (Figure 6b). The mean of 19.76 mM and the standard deviation of 0.53 mM can be calculated from the three concentrations.

3. Some неудачных phrases it is better to re-write. Such as:

Line 12 - nitrogen-doped carbon dots (NCDs) with excitation wavelength,

Line 142 - Tap water was taken from domestic water,

Line 371 - other molecules, unions and cations

Response: Thanks for your kind reminder. These phrases have been corrected.

Line 12-14: “Here, nitrogen-doped carbon dots (NCDs) with excitation and pH dependent optical property, and a high fluorescence quantum yield were successfully synthesized,”

Line 141: “The tap water was used directly without any pre-treatment.”

Line 382-383: “Concentration of OTC and other antibiotics is 20 mM; Concentration of other interfering organic molecules, unions, and cations is 50 mM.”

Thank you again for your considerations!

Sincerely,

Yours,

Email: whf0125@ntu.edu.cn

Nantong Key Laboratory of Intelligent and New Energy Materials, Nantong University, Nantong 226019, Jiangsu, China

Nov. 28th, 2022

Round 2

Reviewer 1 Report

The manuscript has been properly revised.